# Role of Enterprise Alliance in Carbon Emission Reduction Mechanism: An Evolutionary Game Analysis

**DOI:** 10.3390/ijerph191811368

**Published:** 2022-09-09

**Authors:** Jichao Geng, Meiyu Ji, Li Yang

**Affiliations:** School of Economics and Management, Anhui University of Science and Technology, Huainan 232001, China

**Keywords:** carbon tax, carbon trading, enterprise alliance, evolutionary game

## Abstract

This study constructs the enterprise alliance game party, designs the mechanism for which the alliance and the government are jointly responsible for enterprise carbon emission reduction work, and explores the evolutionarily stable strategies (ESSs) of the government, enterprises, and enterprise alliance under the policy of carbon tax and carbon trading with numerical simulations. The results show that: (1) the ESSs of the enterprise alliance are always to give technical support to enterprises; (2) the carbon trading price below the critical value can mobilize the enthusiasm of enterprises for honest emission reduction; (3) the carbon tax rate has a negative correlation with enterprise emission reduction behavior; (4) when the underreported carbon emissions of enterprises exceed the critical value, the enterprise will evolve into dishonest emission reduction. The high carbon emissions underreported by enterprises will mobilize the enthusiasm of the government to choose supervision. This study may be of certain reference significance to optimize the existing carbon emission reduction mechanism and achieve win-win cooperation between enterprises and government in the carbon trading market.

## 1. Introduction

With the development of the economy, energy consumption and carbon emissions continue to rise, triggering environmental pollution and climate warming problems, which have seriously affected the economic activities of enterprises and the daily lives of residents. The development of green finance and the realization of enterprises’ energy savings and emission reduction have become the focus of academic circles [1]. Under the goal of carbon peaking and carbon neutrality, the Central Committee of the Communist Party of China and the State Council in October 2021 clearly emphasized the need to vigorously build a low-carbon recycling development economic system and reduce the level of carbon dioxide emissions. Therefore, the government and relevant departments have successively introduced a series of policies including reward and punishment mechanisms linked to carbon tax and carbon trading, which are important government and market means to achieve the carbon emission reduction of enterprises [2,3]. Enterprises are in the key position of carbon emission reduction [4], but they may choose to dishonestly reduce emissions due to their pursuit of profit maximization, insufficient technical capacity for carbon reduction, and the existence of difficulties in verifying carbon emissions. At this time, the government as a supervisor should actively guide and regulate the emission reduction behavior of enterprises. 

The government’s mechanisms for reducing emissions mainly include rewards and punishments, the carbon trading market, and carbon tax. The academic community has performed much research on the behavioral relationship between the government and enterprises under these mechanisms. Xu et al. [5] introduced government financial interventions and constructed an evolutionary game model between government and enterprises, showing that government regulation can promote the green behavior of enterprises, and suggested that the government should implement the reward and punishment mechanism when the proportion of green supply chain is low. Zhu et al. [6] showed that increasing rewards is not conducive to the government’s regulatory responsibility. Punishment can better improve the convergence rate of enterprises’ strategies compared to rewards in their analysis of the relationship between government and firms in green credit. Niu and Zhu et al. [7] argue that there is no evolutionarily stable strategy in the current Chinese environmental governance system, and the reward and punishment policies of the government only have a positive effect on enterprises’ choices for a short period of time, but a dynamic reward scheme can make the strategy choices of enterprises converge to the ideal state. Huang et al. [8] confirmed that the implementation of a carbon trading mechanism can increase enterprises’ green productivity. Pan et al. [9] explored the mediating role of government participation in the efficiency of the carbon trading market and showed that government regulation significantly moderated the positive effect of the carbon trading market on enterprises’ productivity. Lin et al. [10] showed that carbon trading is effective in curbing carbon emissions, while government intervention plays an important role in carbon emission reduction. Zhu [11] argued that stronger government regulation can encourage enterprises to participate in carbon trading. Fang et al. [12] underlined that excessive government regulation and high carbon emission prices are not conducive to the development of a carbon trading system. Kuo et al. [13] used game theory to study the impact of the government implementation of a carbon tax on enterprises’ technology development and production output and used actual enterprises as examples to show that implementing a carbon tax can achieve energy savings and emission reduction goals. Hu et al. [14] proved that the government’s formulation of a dynamic carbon tax and carbon subsidy mechanisms can better motivate enterprises to reduce emissions. Chen et al. [15] argued that a carbon tax can better motivate enterprises to reduce emissions honestly than a carbon subsidy. Wang et al. [16] pointed out that a heavier carbon tax would make the positive effect of government subsidies more obvious and that government subsidies for emission reduction would bring more social benefits than subsidies for investment costs.

Evolutionary game theory is a powerful tool to study the behavioral game among the subjects involved in carbon finance [17]. Evolutionary game theory assumes that game plays have the characteristics of finite rationality, and game subjects repeatedly play dynamic and repetitive games until they find the optimal stable strategy equilibrium point in the game system [18,19]. Most researchers have used two-party or three-party game players, of which two-party game players consist mainly of government and enterprises [20], financial institutions and users [21], and enterprises and the public [22]. For example, Zhang et al. [23] constructed an evolutionary game framework for supply chain enterprises and analyzed the evolution path of enterprises’ green behavior strategies, indicating that enterprises need to reduce green investment costs and participate in collaborative management. Among the three-party games, Xie et al. [24] constructed an evolutionary game of supply chain enterprises about manufacturers, suppliers, and distributors to find evolutionarily stable strategies (ESSs) and promote collaborative innovation among enterprises. Su et al. [25] argued that building a core enterprise-oriented entrepreneurship ecosystem has an important role in promoting sustainable development and realizing the multi-agent win-win cooperation. Zhu et al. [26] explored the relationship between the central government, local government, and enterprises by using evolutionary game theory and developed an evolutionarily stable strategy to ensure the effective implementation of fiscal policy. Wang et al. [27] suggested that the central and local governments jointly improve the carbon tax and reward and punishment mechanisms. Chen and Zhang et al. [28] studied the relationship between government, enterprises, and the public and showed that public participation in governance can replace government supervision to a certain extent. Zou et al. [29] believe that public participation facilitates the government to perform its supervisory role, while the government formulating effective policies can promote the research and development of low-carbon technologies in enterprises. Achieving energy conservation and emission reduction requires enterprises to enhance carbon reduction technologies, and Qu et al. [30] showed that improving carbon tax pricing can promote low-carbon technology innovation. Wang et al. [31] studied low-carbon technology innovation strategies with cooperation among government, universities, and enterprises under a carbon trading policy and showed that changing government subsidy factors and carbon trading prices can significantly increase the optimal level of effort of traditional enterprises. Wu et al. [32] showed that the decision types between enterprises do not affect the level of low-carbon technology innovation, while the emission strategies of supplying players achieve Pareto optimality through a cost-sharing contract.

In summary, it can be seen that the literature has used the evolutionary game to find the replicated dynamic equations based on a series of governmental emission reduction policies, focusing on the evolutionary game behaviors of the government, the enterprises, and the public [33,34]. However, relevant studies often select a certain type of carbon trading, reward, punishment, or carbon tax when formulating governmental emission reduction strategies; the combined effect of several emission reduction mechanisms has not been comprehensively considered. Moreover, the definition of the carbon tax is relatively limited, and the carbon tax is only levied on enterprises with excess carbon emissions. Most importantly, the existing literature considers that in the carbon emission reduction mechanism, the research subjects are the government and enterprise, and the government plays the core role, which is responsible for the setting of the carbon price in the carbon trading market, the allocation of the carbon quota, the supervision of the enterprises, the rewards and punishments, etc. However, the achievement of the carbon peaking and carbon neutrality goals requires the key guarantee of enterprises’ innovation and cooperation. Therefore, the fundamental point of government regulation is how to promote the positive interaction and cooperation among enterprises in technological innovation, accelerate the speed of technological innovation in the industry, and reduce the cost of carbon reduction. It is urgent for the government to innovate the way of supervision, build a win-win cooperation mechanism between government and enterprises, and stimulate the innovation vitality of enterprises. 

Under the development concept of coordination, openness, and sharing, enterprise alliance has become an important enterprise cooperation behavior. Enterprise alliance is conducive to finding the resources required by the global market, reducing the risk of innovation and input costs for enterprises in the alliance, as well as reducing unnecessary government intervention in the market and bringing into play the vitality of market players. Therefore, this paper introduces a new game player—enterprise alliance—which is responsible for the carbon quota of enterprises and assumes the emission reduction responsibility of all enterprises under it. Unlike the carbon market constructed by the government, the alliance itself has the nature of a carbon market. Enterprises that join the alliance can trade within the alliance. When the alliance chooses to provide low-carbon technology support to enterprises, enterprises can buy and sell at a lower carbon price in the internal carbon trading market. The existence of the alliance brings potential vitality to enterprises’ independent emission reduction behavior, which is conducive to reducing the cost of emission reduction and carbon price for enterprises and industries, thus increasing their willingness to reduce emissions. Enterprise alliance contributes to the development of a low-carbon economy while easing the pressure on the government to develop the carbon trading market. 

Based on this, this paper constructs an evolutionary game model of the government, enterprises, and enterprise alliances under the carbon tax, carbon trading, and government reward and punishment mechanisms and strictly defines the carbon tax as the product of the carbon tax rate and the carbon emissions of enterprises. The evolutionarily stable strategies of the three parties with and without the carbon tax and carbon trading mechanisms are studied, respectively, to provide a reference for finding the optimal path to carbon emission reduction. This paper contributes mainly in the following aspects:(1)The new game player of enterprise alliance is constructed, and the alliance helps the positive interaction and cooperation of technological innovation among enterprises and reduces the carbon reduction cost of enterprises.(2)The alliance and the government are jointly responsible for carbon emission reduction, which further optimizes the government’s regulatory model.(3)The impact of the carbon trading price, carbon tax, and other variables on enterprises’ carbon emission reduction behavior is proposed to provide relevant policy suggestions for the government and enterprises.

The structure of this paper is as follows: Section 2 constructs a game model of the three parties involved in carbon emission reduction. In Section 3, three different emission reduction mechanism scenarios are formulated to explore the evolutionarily stable strategies of the three parties with and without the carbon tax and carbon trading policies. Section 4 gives the numerical simulation results and analyzes the effects of key variables on the strategies of the game players. Section 5 presents the conclusions and Section 6 presents the related policy recommendations. 

## 2. Evolutionary Game Model Construction

### 2.1. Model Assumptions

First, assume that enterprises, enterprise alliance, and the government are the three parties in the game. The enterprise is a member of the alliance, which is in charge of the enterprises and responsible for them. The government is responsible for the alliance directly as the strategy maker. The government gives a fixed carbon quota to the alliance, which is responsible for allocating it to each enterprise. Assume that the number of enterprises under the alliance is N and the average carbon quota received by each enterprise is e_0_. As emission reduction agents, enterprises can trade in the carbon market and choose to reduce emissions honestly or dishonestly. Honest emission reduction means carbon trading based on their real carbon emissions. Dishonesty means misrepresenting their carbon emissions (generally considered as underrepresentation) and trading on the carbon market with the misrepresented carbon emissions. The alliance is responsible for the enterprises and can choose to give them technical support or not. Giving technical support to enterprises will bring some costs to the alliance, but it will be beneficial to the enterprises’ profits, which will be reflected in the reduction of their emission reduction costs and the price of carbon trading. As a regulator and policy-maker, the government supervises the emission reduction behavior of enterprises by checking their carbon emission reports and other measures. Supervision will generate supervision costs, but it is beneficial to detect the dishonest behavior of enterprises, disciplining industry and enterprises’ behavior through rewards and punishments and adjusting carbon trading prices. The assumptions of the variables involved in the tripartite game are as follows:

**Hypothesis** **1.**
*It is assumed that the probability of enterprises choosing honest emission reduction is x (0 ≤ x ≤ 1), and the probability of dishonesty is 1 − x; the probability of the alliance choosing to give technical support to enterprises is y (0 ≤ y ≤ 1), and the probability of not supporting is 1 − y; the probability of the government choosing to supervise is z (0 ≤ z ≤ 1), and the probability of not supervising is 1 − z.*


**Hypothesis** **2.**
*Enterprises choosing to reduce emissions honestly will bring positive market benefits, and the market income of R_1_ at this time is greater than the market income of R_2_ when enterprises reduce emissions dishonestly. When the alliance chooses technical support, enterprises obtain the reward of J issued by the government to the alliance with a proportion of 1 − α, and the cost of reducing emissions honestly C_1_ is greater than the cost C_3_ when enterprises are dishonest. The profits of enterprises in the carbon trading market are related to their carbon emissions as P (e_0_ − e_i_), which can be positive or negative, and e_i_ represents the carbon emissions in the enterprise’s carbon report. If the government finds that the enterprise is dishonest, it will punish the fraction of misreported carbon emissions with a penalty coefficient of γ. When the government sets a carbon tax policy and the tax rate is µ, the enterprise pays a carbon tax of µe_i_.*


**Hypothesis** **3.**
*The alliance will increase the cost of C_0_ for technical support to enterprises and obtain the government reward of αJ, αJ > C_0_ and obtain fixed income S_i_. When enterprises are honest, the fixed income S_1_ obtained by the alliance with technical support is greater than the fixed income S_3_ obtained without support. The alliance obtains additional rewards or punishments from the government related to the excess carbon quota of enterprises, and the coefficient of rewards and punishments is β. The alliance obtains rewards or punishments with Nβ(e_0_ − e_i_).*


**Hypothesis** **4.***The cost incurred by the government when it chooses to supervise the enterprise is C. If the supervision finds that the enterprise has dishonest emission reduction behavior, it will be fined with γ(e_1_ − e_3_) or γ(e_2_ − e_4_). When the alliance gives technical support and the enterprises are honest in reducing emissions, the government gains green benefits H_1_, including the benefits of mitigating environmental pollution and saving energy consumption, while the government’s green benefits are reduced when enterprises are dishonest in reducing emissions or the alliance does not give the enterprises technical support*.

The parameters of interest for the three parties of the game are defined in Table 1.

### 2.2. Model Construction

Based on the above description of the three parties of the game and the associated assumptions, the game payoff matrix is shown in Table 2.

## 3. Evolutionary Game Analysis

### 3.1. Strategy Stability Analysis for Enterprises 

Suppose that the expected profit when the enterprise chooses to reduce emissions honestly is denoted as $U_{11}$, the expected profit when it chooses to reduce emissions dishonestly is denoted as $U_{12}$, and the average expected profit of the enterprise is $\bar{U_{1}}$; we have:(1)$$U_{11}=R_{1}-C_{2}-P_{2}\left(e_{2}-e_{0}\right)-\mu e_{2}+y\left[C_{2}-C_{1}+P_{2}\left(e_{2}-e_{0}\right)-P_{1}\left(e_{1}-e_{0}\right)+\mu \left(e_{2}-e_{1}\right)+\left(1-\alpha \right)J\right]$$
(2)$$U_{12}=R_{2}-C_{4}-P_{2}\left(e_{4}-e_{0}\right)-\mu e_{4}-z\gamma \left(e_{2}-e_{4}\right)+y[C_{4}-C_{3}+P_{2}\left(e_{4}-e_{0}\right)-P_{1}\left(e_{3}-e_{0}\right)\;+\mu \left(e_{4}-e_{3}\right)+\left(1-\alpha \right)J]+yz\gamma \left(e_{2}-e_{4}+e_{3}-e_{1}\right)\;$$
(3)$${\bar{U}}_{1}=xU_{11}+\left(1-x\right)U_{12}$$

The replication dynamic equation of the enterprise is: (4)$$L\left(x\right)=\frac{dx}{dt}=x\left(U_{11}-{\bar{U}}_{1}\right)=x\left(1-x\right)\left(U_{11}-U_{12}\right)=x\left(1-x\right)D_{1}\;=x\left(1-x\right)\{R_{1}-R_{2}+C_{4}-C_{2}+P_{2}\left(e_{4}-e_{2}\right)+\mu \left(e_{4}-e_{2}\right)+z\gamma \left(e_{2}-e_{4}\right)+yz\gamma \left(e_{1}-e_{3}+e_{4}-e_{2}\right)+y[c_{3}-c_{1}\;+c_{2}-c_{4}+P_{2}\left(e_{2}-e_{4}\right)+P_{1}\left(e_{3}-e_{1}\right)+\mu \left(e_{3}-e_{1}+e_{2}-e_{4}\right)]\}$$

The derivative of L(x) yields: $\frac{dL\left(x\right)}{dx}=\left(1-2x\right)D_{1}$, where $D_{1}$ is a function on *y*, *z*. When $D_{1}=0$, $y^{*}=\frac{-\left[R_{1}-R_{2}+C_{4}-C_{2}+P_{2}\left(e_{4}-e_{2}\right)+\mu \left(e_{4}-e_{2}\right)\right]-z\gamma \left(e_{2}-e_{4}\right)}{\left[C_{3}-C_{1}+C_{2}-C_{4}+P_{2}\left(e_{2}-e_{4}\right)+P_{1}\left(e_{3}-e_{1}\right)+\mu \left(e_{3}-e_{1}+e_{2}-e_{4}\right)\right]+z\gamma \left(e_{1}-e_{3}+e_{4}-e_{2}\right)}$, at this time L(x)=0, no matter what value the probability *x* of honest emission reduction is, the choice of the enterprise belongs to the ESS. According to the differential equation stability theorem, the ESS of the enterprise needs to satisfy the conditions: L(x)=0, $\frac{dL\left(x\right)}{dx}<0$, so only *x* takes the value of 0 or 1. The details are shown in Figure 1.

Figure 1 shows that when the probability of the enterprise alliance choosing to give technical support to enterprises is higher than a certain level, *x* = 1 is the stable strategy point for the enterprise, when the enterprise prefers to reduce emissions honestly, while less than a certain level, *x* = 0 is the stable strategy point for the enterprise, and at this time, the enterprise will shift to dishonest emission reduction.

### 3.2. Strategy Stability Analysis for Enterprise Alliance 

It is assumed that the expected profit of an enterprise alliance when it chooses to give technical support to enterprises is denoted as $U_{21}$, the expected profit when it chooses not to give technical support is denoted as $U_{22}$, and the average expected profit of the enterprise alliance is $\bar{U_{2}}$; we have:(5)$$U_{21}=\alpha J-C_{0}+S_{2}+N\beta \left(e_{0}-e_{3}\right)+x\left[S_{1}-S_{2}+N\beta \left(e_{3}-e_{1}\right)\right]$$
(6)$$U_{22}=S_{4}+N\beta \left(e_{0}-e_{4}\right)+x\left[S_{3}-S_{4}+N\beta \left(e_{4}-e_{2}\right)\right]$$
(7)$${\bar{U}}_{2}=yU_{21}+\left(1-y\right)U_{22}$$

The replication dynamic equation for the enterprise alliance is: (8)$$M\left(y\right)=\frac{dy}{dt}=y\left(U_{21}-{\bar{U}}_{2}\right)=y\left(1-y\right)\left(U_{21}-U_{22}\right)=y\left(1-y\right)D_{2}\;=y\left(1-y\right)\left\{\alpha J-C_{0}+S_{2}-S_{4}+N\beta \left(e_{4}-e_{3}\right)+x\left[S_{1}-S_{3}+S_{4}-S_{2}+N\beta \left(e_{3}-e_{1}+e_{2}-e_{4}\right)\right]\right\}$$

The derivative of M(y) yields $\frac{dM\left(y\right)}{dy}=\left(1-2y\right)D_{2}$, where $D_{2}$ is a function on *x*. When $D_{2}=0$, $x^{*}=\frac{-\left[\alpha J-C_{0}+S_{2}-S_{4}+N\beta \left(e_{4}-e_{3}\right)\right]}{S_{1}-S_{3}+S_{4}-S_{2}+N\beta \left(e_{3}-e_{1}+e_{2}-e_{4}\right)}$, at this time, M(y)=0. Regardless of the value of the probability *y* that the enterprise alliance chooses to give the enterprise technical support, the choice of the enterprise alliance belongs to the ESS. According to the differential equation stability theorem, the ESS of the alliance needs to satisfy the conditions: M(y)=0, $\frac{dM\left(y\right)}{dy}<0$, so only *y* takes the value of 0 or 1. The details are shown in Figure 2.

Figure 2 shows that when the probability of enterprises choosing honest emission reduction is higher than a certain level, *y* = 1 is the stable strategy point of the alliance, and the alliance is more willing to give technical support to enterprises at this time, while less than the certain level, *y* = 0 is the stable strategy point of the alliance.

### 3.3. Strategy Stability Analysis for Government

Assuming that the expected profit when the government chooses to supervise the enterprises’ emission reduction is denoted as $U_{31}$, the expected profit when it chooses not to supervise is denoted as $U_{32}$, and the government’s average expected profit is $\bar{U_{3}}$, we have:(9)$$U_{31}=H_{2}-C-L+\mu e_{4}+\gamma \left(e_{2}-e_{4}\right)-N\beta \left(e_{0}-e_{4}\right)+x[\left(\mu -\gamma +N\beta \right)\left(e_{2}-e_{4}\right)+L]+y[H_{1}-H_{2}-J+(\mu \;+N\beta )\left(e_{3}-e_{4}\right)+\gamma \left(e_{1}-e_{3}+e_{4}-e_{2}\right)]+xy\left(\mu -\gamma +N\beta \right)\left(e_{1}-e_{3}+e_{4}-e_{2}\right)$$
(10)$$U_{32}=H_{2}-L+\mu e_{4}-N\beta \left(e_{0}-e_{4}\right)+x[\left(\mu +N\beta \right)\left(e_{2}-e_{4}\right)+L]+y[H_{1}-H_{2}-J+\left(\mu +N\beta \right)(e_{3}\;-e_{4})]+xy\left(\mu +N\beta \right)\left(e_{1}-e_{3}+e_{4}-e_{2}\right)$$
(11)$${\bar{U}}_{3}=zU_{31}+\left(1-z\right)U_{32}$$

The replication dynamic equation for the government is:(12)$$N\left(z\right)=\frac{dz}{dt}=z\left(U_{31}-{\bar{U}}_{3}\right)=z\left(1-z\right)\left(U_{31}-U_{32}\right)=z\left(1-z\right)D_{3}\;\;=z\left(1-z\right)\left\{-C+y\left(1-x\right)\gamma \left(e_{1}-e_{3}\right)+\left(1-y\right)\left(1-x\right)\gamma \left(e_{2}-e_{4}\right)\right\}$$

The derivative of N(z) yields $\frac{dN\left(z\right)}{dz}=\left(1-2z\right)D_{3}$, where $D_{3}$ is a function on *x*, *y*. When $D_{3}=0$, $y^{**}=\frac{-\left[-C+\gamma \left(e_{2}-e_{4}\right)-x\gamma \left(e_{2}-e_{4}\right)\right]}{\gamma \left(e_{1}-e_{3}+e_{4}-e_{2}\right)-x\gamma \left(e_{1}-e_{3}+e_{4}-e_{2}\right)}$, at which time N(z)=0, the government’s choice belongs to the ESS, regardless of the value of the probability *z* that the government chooses to supervise. According to the differential equation stability theorem, the government’s ESS needs to satisfy the conditions: N(z)=0, $\frac{dN\left(z\right)}{dz}<0$, so only *z* takes the value of 0 or 1. The details are shown in Figure 3.

Figure 3 shows that when the probability of the alliance choosing to give technical support to enterprises is higher than a certain level, *z* = 1 is the government’s stabilization strategy point, when the government is more willing to supervise the enterprises’ emission reduction behavior, while less than the certain level, *z* = 0 is the government’s stabilization strategy point.

### 3.4. Stability Analysis of the Equilibrium Point of the Three-Party Game System

On the basis of the analysis of the strategy stability of the individual game player, the three-party game system is analyzed as a whole. Let $\frac{dx}{dt}=0$, $\frac{dy}{dt}=0$, $\frac{dz}{dt}=0$; the equilibrium points of the three-party game system can be obtained: E_1_ (0, 0, 0), E_2_ (0, 1, 0), E_3_ (0, 0, 1), E_4_ (0, 1, 1), E_5_ (1, 0, 0), E_6_ (1, 1, 0), E_7_ (1, 0, 1), E_8_ (1, 1, 1), E_9_ ($\frac{\gamma \left(e_{2}-e_{4}\right)-C}{\gamma \left(e_{2}-e_{4}\right)}$, 0, $\frac{R_{2}-R_{1}+C_{2}-C_{4}+\left(P_{2}+\mu \right)\left(e_{2}-e_{4}\right)}{\gamma \left(e_{2}-e_{4}\right)}$), E_10_ ($\frac{\gamma \left(e_{1}-e_{3}\right)-C}{\gamma \left(e_{1}-e_{3}\right)}$, 1, $\frac{R_{2}-R_{1}+C_{1}-C_{3}+\left(P_{1}+\mu \right)\left(e_{1}-e_{3}\right)}{\gamma \left(e_{1}-e_{3}\right)}$), E_11_ ($\frac{C_{0}-J\alpha +S_{4}-S_{2}+N\beta \left(e_{3}-e_{4}\right)}{S_{1}-S_{3}+S_{4}-S_{2}+N\beta \left(e_{3}-e_{1}+e_{2}-e_{4}\right)}$, $\frac{R_{2}-R_{1}+C_{2}-C_{4}+\left(P_{2}+\mu \right)\left(e_{2}-e_{4}\right)}{C_{3}-C_{1}+C_{2}-C_{4}+P_{2}\left(e_{2}-e_{4}\right)+P_{1}\left(e_{1}-e_{3}\right)+\mu \left(e_{3}-e_{1}+e_{2}-e_{4}\right)}$, 0), E_12_ ($\frac{C_{0}-J\alpha +S_{4}-S_{2}+N\beta \left(e_{3}-e_{4}\right)}{S_{1}-S_{3}+S_{4}-S_{2}+N\beta \left(e_{3}-e_{1}+e_{2}-e_{4}\right)}$, $\frac{R_{2}-R_{1}+C_{2}-C_{4}+\left(P_{2}+\mu -\gamma \right)\left(e_{2}-e_{4}\right)}{C_{3}-C_{1}+C_{2}-C_{4}+P_{2}\left(e_{2}-e_{4}\right)+P_{1}\left(e_{1}-e_{3}\right)+\left(\mu -\gamma \right)\left(e_{3}-e_{1}+e_{2}-e_{4}\right)}$, 1), E_13_ (*x*_0_, *y*_0_, *z*_0_).

#### 3.4.1. The Jacobian Matrix and Its Eigenvalues

According to the method proposed by Friedman [35], the stability of the equilibrium points can be analyzed by the Jacobian matrix of this game system. From Lyapunov’s indirect method [36], we know that if all three eigenvalues of the Jacobian matrix have negative real parts, then the equilibrium has stability. The Jacobian matrix of the game system of enterprises, enterprise alliance, and the government can be represented by *J*. The partial derivatives of *L*(*x*), *M*(*y*), and *N*(*z*) with respect to *x*, *y*, and *z* are obtained as *J*:(13)$$J=[\begin{array}{ccc} \left(1-2x\right)D_{1} & \left(x-x^{2}\right)[z\gamma \left(e_{1}-e_{3}+e_{4}-e_{2}\right)+C_{3}-C_{1}+C_{2}-\;C_{4}+P_{2}\left(e_{2}-e_{4}\right)+P_{1}\left(e_{3}-e_{1}\right)+\mu \left(e_{3}-e_{1}+e_{2}-e_{4}\right)] & \left(x-x^{2}\right)[\gamma \left(e_{2}-e_{4}\right)\;y\gamma \left(e_{1}-e_{3}+e_{4}-e_{2}\right)]\\ \left(y-y^{2}\right)[S_{1}-S_{3}+S_{4}-S_{2}\;+N\beta \left(e_{3}-e_{1}+e_{2}-e_{4}\right)] & \;\;\left(1-2y\right)D_{2} & \;\;0\\ \left(z-z^{2}\right)[-\gamma \left(e_{2}-e_{4}\right)-\;y\gamma \left(e_{1}-e_{3}+e_{4}-e_{2}\right)] & \left(z-z^{2}\right)[\gamma \left(e_{1}-e_{3}+e_{4}-e_{2}\right)\;-x\gamma \left(e_{1}-e_{3}+e_{4}-e_{2}\right)] & \;\;\left(1-2z\right)D_{3} \end{array}]$$

By calculating the Jacobian matrix eigenvalues, the equilibrium points E_9_ to E_13_ do not satisfy the condition that all three eigenvalues have negative real parts, and the matrix eigenvalues of the remaining eight equilibrium points are shown in Table 3.

Considering several situations where the government has or does not implement a carbon tax on enterprises and has or does not have a carbon trading market, the following three mechanisms are discussed in detail. The calculations show that the matrix eigenvalues *λ*_2_ and *λ*_3_ are the same under the three mechanisms. Regardless of whether enterprises choose to reduce emissions honestly or dishonestly, the fixed benefits brought by the alliance’s choice of technical support are greater than those without technical support; therefore, *S*_1_ − *S*_3_ > 0 and *S*_2_ − *S*_4_ > 0. Technical support for enterprises from the alliance will lead to the reduction of the overall carbon emissions of enterprises, so *Nβ*(*e_4_* − *e*_3_) > 0 and *Nβ*(*e*_2_ − *e*_1_) > 0, with *αJ* − *C*_0_*+ S*_2_ − *S*_4_
*+ Nβ*(*e*_4_ − *e*_3_) > 0 and *αJ* − *C*_0_ *+ S*_2_ − *S*_4_ *+ Nβ*(*e*_2_ − *e*_1_) > 0. From this, we can know the positive or negative eigenvalue λ_2_ of each equilibrium point. The stability of equilibrium points under the mechanism of the carbon tax with carbon trading (*P* ≠ 0, *µ* ≠ 0), the carbon tax without carbon trading (*P*_1_ = 0, *P*_2_ = 0), and carbon trading without a carbon tax (*µ* = 0) is shown in the following Table 4.

#### 3.4.2. Discussion of ESS under Three Mechanisms

From the table, it can be seen that the three-party game system replicating the dynamic equation tends to stabilize mainly at the points (0, 1, 0), (0, 1, 1), and (1, 1, 0) under different mechanisms, and the positive or negative eigenvalues *λ*_1_ and *λ*_3_ under the three mechanisms are discussed below, so as to determine the ESS under each mechanism:(1)With carbon tax and carbon trading (*P* ≠ 0, *µ* ≠ 0):

When *γ*(*e*_1_ − *e*_3_) − *C* > 0, *R*_1_ − *R*_2_ + *C*_3_ − *C*_1_ + *P*_1_(*e*_3_ − *e*_1_) + (*µ* − *γ*)(*e*_3_ − *e*_1_) < 0, the government’s punishment for enterprises’ misreporting of carbon emissions is greater than its own supervision cost and the net profit from honest emission reduction by enterprises is smaller than the net profit of dishonest emission reduction when the alliance gives technical support to enterprises and the government chooses to supervise. At this point, the Jacobian matrix eigenvalues at the equilibrium point (0, 1, 1) are all negative, and the equilibrium point is an ESS with a combination of the strategy for enterprises to choose dishonesty, the alliance to choose technical support, and the government to choose supervision.

When *γ*(*e*_1_ − *e*_3_) − *C* < 0 and *R*_1_ − *R*_2_ + *C*_3_ − *C*_1_ + *P*_1_(*e*_3_ − *e*_1_) + *µ*(*e*_3_ − *e*_1_) < 0, the government’s punishment for enterprises’ misreporting of carbon emissions is smaller than its own supervision cost and the net profit brought by enterprises’ honest emission reduction is smaller than the net profit of dishonest emission reduction when the alliance gives technical support to enterprises and the government does not supervise. At this time, the Jacobian matrix eigenvalues at the equilibrium point (0, 1, 0) are all negative, and the equilibrium point is an ESS with a combination of the strategy for enterprises to choose dishonesty, the alliance to choose technical support, and the government to choose no supervision.

When −[*R*_1_ − *R*_2_ + *C*_3_ − *C*_1_ + *P*_1_(*e*_3_ − *e*_1_) + *µ*(*e*_3_ − *e*_1_)] < 0, that is, in the case where the alliance gives technical support to enterprises and the government does not supervise, the net profit brought by the enterprise’s honest emission reduction is greater than the net profit of the dishonest emission reduction. At this point, the Jacobian matrix eigenvalues of the equilibrium point (1, 1, 0) are all negative, and the equilibrium point is an ESS with the strategy combination of enterprises choosing honesty, the alliance choosing technical support, and the government choosing no supervision.

(2)With carbon tax and no carbon trading (*P*_1_ = 0, *P*_2_ = 0):

When *γ*(*e*_1_ − *e*_3_) − *C* > 0 and *R*_1_ − *R*_2_ + *C*_3_ − *C*_1_ + (*µ* − *γ*)(*e*_3_ − *e*_1_) < 0, the equilibrium point (0, 1, 1) is an ESS, and the strategy combination is that enterprises choose dishonesty, the alliance chooses technical support, and the government chooses supervision.

When *γ*(*e*_1_ − *e*_3_) − *C* < 0 and *R*_1_ − *R*_2_ + *C*_3_ − *C*_1_ + *µ*(*e*_3_ − *e*_1_) < 0, the equilibrium point (0, 1, 0) is the ESS, and the strategy combination is that the enterprises choose dishonesty, the alliance chooses technical support, and the government chooses no supervision.

When −[*R*_1_ − *R*_2_ + *C*_3_ − *C*_1_ + *µ*(*e*_3_ − *e*_1_)] < 0, the equilibrium point (1, 1, 0) is the ESS, and the strategy combination is that the enterprises choose honesty, the alliance chooses technical support, and the government chooses not to supervise.

(3)No carbon tax, with carbon trading (*µ* = 0):

When *γ*(*e*_1_ − *e*_3_) − *C* > 0, *R*_1_ − *R*_2_ + *C*_3_ − *C*_1_ + *P*_1_(*e*_3_ − *e*_1_) − *γ*(*e*_3_ − *e*_1_) < 0, the equilibrium point (0, 1, 1) is the ESS, and the strategy combination is that the enterprises choose dishonesty, the alliance chooses technical support, and the government chooses supervision.

When *γ*(*e*_1_ − *e*_3_) − *C* < 0 and *R*_1_ − *R*_2_ + *C*_3_ − *C*_1_ + *P*_1_(*e*_3_ − *e*_1_) < 0, the equilibrium point (0, 1, 0) is the ESS, and the strategy combination is that the enterprises choose dishonesty, the alliance chooses technical support, and the government chooses no supervision.

When −[*R*_1_ − *R*_2_ + *C*_3_ − *C*_1_ + *P*_1_(*e*_3_ − *e*_1_)] < 0, the equilibrium point (1, 1, 0) is the ESS, and the strategy combination is that the enterprises choose honesty, the alliance chooses technical support, and the government chooses no supervision.

Combining the above analysis, the conditions that need to be satisfied at each stabilization point under the three mechanisms can be calculated. For the convenience of analysis, *e*_1_ − *e*_3_ involved in this table is replaced by Δ*e* (Δ*e* = *e*_1_ − *e*_3_), as shown in Table 5.

## 4. Numerical Experiments and Simulations

This paper used numerical simulation to further validate the above constructed evolutionary game model. Set the variable values according to the condition $P_{1}+\mu <\frac{R_{1}-C_{1}-R_{2}+C_{3}}{\Delta e}$ that satisfies the stabilization strategy of (1, 1, 0). Referring to the related literature [37], let *R*_1_ = 130, *R*_2_ = 100, *C*_1_ = 55, *C*_2_ = 60, *C*_3_ = 50, *C*_4_ = 55, *P*_1_ = 8, *P*_2_ = 10, *e*_0_ = 50, *e*_1_ = 50, *e*_2_ = 52, *e*_3_ = 48, *e*_4_ = 50, *µ* = 4, *J* = 50, *α* = 0.7, *γ* = 10, *C*_0_ = 20, *S*_1_ = 30, *S*_2_ = 28, *S*_3_ = 26, *S*_4_ = 24, *N* = 100, *β* = 2, *C* = 30, *H*_1_ = 20, *H*_2_ = 15, *L* = −10, and numerical simulations of various scenarios were performed using MATLAB software. Figure 4 gives the evolutionary strategies diagram of the three parties of the game with different initial wills, and the image shows that no matter the initial will of *x*, *y*, and *z* being 0.4, 0.5 or 0.6, the final evolutionary result is the ideal state of *x*→1, *y*→1 and *z*→0.

Assuming that the willingness of all three players of the game is 0.5, next, we simulated different carbon trading prices, carbon tax rates, and the underreported carbon emissions when enterprises are dishonest to observe the effects of different variables on the choice of game players.

### 4.1. The Effect of Carbon Trading Price on the Evolutionary Game

The enterprise alliance itself has the nature of a carbon market, and the enterprises within the alliance can enjoy a preferential carbon trading price of *P*_1_ compared with the market. Taking the value of *P*_1_ from 6 to 9 while keeping the values of other variables constant, we observed the change of the choice probability of the game players under different values of *P*_1_.

From Figure 5a, when *P*_1_ < 8.5, the probability of enterprises choosing honest emission reduction evolves to *x* = 1, the enterprise will eventually choose honest emission reduction, and the probability of choosing honest emission reduction will tend to 1 faster as the carbon trading price decreases. When *P*_1_ = 8.5, the evolution result is (0.65, 1, 0), and the probability of enterprises choosing honest emission reduction is 65% at this time. When the carbon trading price of *P*_1_ is 9, enterprises will choose dishonest emission reduction. It can be found that enterprises choose honest emission reduction behavior related to the carbon trading price, and there is a critical point of *P*_1_, which is between 8.5 and 9. When the price of carbon trading within the alliance is less than this critical point, enterprises choose to reduce emissions honestly, while beyond this critical point, enterprises will shift from honest emission reduction to dishonest. A price near the critical point means that the price of carbon trading within the alliance is close to the market price, and a small change in the price of carbon trading will cause a large change in the behavior of enterprises. Therefore, the alliance needs to control the carbon trading price to fluctuate within a certain range to ensure the effectiveness of the alliance’s technical support to enterprises. At the same time, Figure 5a shows that, in general, the carbon trading price does not have much influence on the government’s evolutionary result, and eventually, the government will choose not to supervise. Different carbon trading prices affect the speed at which the government tends to choose not to supervise; the lower the price, the faster the probability that the government will choose not to supervise tends to 0. From Figure 5b, it can be seen that the carbon trading price basically has no impact on the choice of enterprise alliance.

### 4.2. The Impact of Carbon Tax Rate on the Evolutionary Game

Carbon tax is one of the means for the government to regulate enterprises’ emission reduction behavior. As can be seen from Figure 6a, with the change of the carbon tax rate, the probability that enterprises choose to reduce emissions honestly also changes. When the carbon tax rate *μ* < 5, the probability of enterprises’ honest emission reduction eventually evolves to 1; when *μ* > 4, enterprises shift from honest emission reduction to dishonesty. Therefore, there is also a critical point for the carbon tax rate, which lies between 4 and 5. Combined with Figure 6b, the stabilization point (1, 1, 0) turns into (0, 1, 0) when the tax rate exceeds this critical point. Further observation of Figure 6c shows that when the carbon tax rate is less than this critical point, it takes longer for enterprises to reach the probability of honesty of 1 as the price rises; beyond that critical point, the higher the carbon tax rate, the faster enterprises tend to reduce emissions dishonestly. The government can effectively control the emission reduction behavior of enterprises by adjusting the size of the carbon tax rate. To achieve the goal of honest emission reduction by enterprises, the government should set the condition that the tax rate is less than the critical point, and the size of the carbon tax rate can be adjusted according to the speed of achieving the goal.

### 4.3. Impact of Enterprises’ Underreported Carbon Emissions on the Evolutionary Game

When an enterprise chooses to reduce emissions dishonestly, it will misreport a lower amount of carbon emissions than if it reduced emissions honestly, and the amount of understatement varies due to enterprise differences. From Figure 7a–c, it can be seen that changing the amount of underreported carbon emissions when enterprises are dishonest has an impact on the choice of game players. Specifically, when Δ*e* = 2, the game evolves as *x* → 1, *y* → 1, *z* → 0; when 2 < Δ*e* < 12, the choice probability of enterprises and government oscillates up and down in a specific interval with the passage of time; when Δ*e* > 12, the game evolves as *x* → 0, *y* → 1, *z* → 1. It can be seen that for government supervision and enterprise emission reduction behavior, there is a critical point for enterprises’ underreported carbon emissions. The first critical point of underreporting carbon emissions is around 2, when enterprises shift from honesty to dishonesty; the second critical point is around 12, and the government has shifted from non-supervision to supervision. Combined with Figure 7d, we know that the underreporting of carbon emissions when enterprises are dishonest has a greater impact on the evolutionary results.

## 5. Conclusions

This paper analyzed the evolutionary results of the three game players between enterprises, enterprise alliance, and the government under a series of policies of carbon tax formulated by the government and carbon trading implemented within enterprise alliance. The influence of the carbon trading price, carbon tax rate, and carbon emissions underreported by dishonest enterprises on the evolution results was discussed. The main findings of the study are as follows:

(1)The evolutionary results of the three game players of enterprises, business alliance, and governments are not uniquely invariant; the carbon trading price, carbon tax rate, and the underreported carbon emissions when enterprises are dishonest all have an impact on the ESS. Changes in parameters lead to shifts in different behaviors of enterprises and the government. However, no matter how these parameters change, the choice of enterprise alliance is always to give technical support to enterprises. This stable strategic choice of enterprise alliance can effectively promote the healthy exchange of technological innovation among enterprises, accelerate the speed of industrial technological innovation, and reduce the cost of carbon reduction.(2)For enterprises’ emission reduction behavior, there is a critical point of carbon trading price within the alliance, which is between 8.5 and 9. When the price of carbon trading is lower than this critical point, enterprises will actively choose to reduce emissions honestly, and the technical support given to enterprises by the alliance will only be effective at this time. As the price of carbon trading rises, the enthusiasm of enterprises will decline; when the price rises close to the market price, a small change in the price of carbon trading will cause a greater degree of behavioral change in enterprises, and eventually, enterprises will shift from honest emission reduction to dishonest.(3)The level of the carbon tax rate affects the emission reduction behavior of enterprises. When the carbon tax rate exceeds the critical point (between 4 and 5), enterprises tend to reduce emissions dishonestly, and the higher the carbon tax rate, the faster enterprises tend to reduce emissions dishonestly. Controlling the carbon tax rate will make the game result evolve towards an ideal state (that is, the stability strategy of enterprise honesty, alliance support, and government non-supervision).(4)Underreporting of carbon emissions when enterprises are dishonest affects the behavior of both the government and enterprises. If an enterprise is skilled enough to reduce emissions, the carbon emissions underreported when it chooses to be dishonest are small. In this case, the government will not supervise the enterprise, and most enterprises will eventually choose to be honest in emission reduction over time. If an enterprise underreports a large amount of carbon emissions when it is dishonest, the government will actively supervise the behavior of the enterprise. However, the enterprise will choose to reduce emissions dishonestly in the end, which may be caused by the insufficient technical ability of the enterprise or the “herd effect” of the dishonest behavior of enterprises in the alliance.

## 6. Policy Implications

Based on the above findings, the following recommendations are made from the perspectives of the government, alliance, and enterprises, respectively:

(1)We suggest the government weaken some of its regulatory functions over enterprises and allocate some of the power to enterprise alliances, which are responsible for the internal management of enterprises including carbon quotas. At the same time, the government should require the enterprise alliance to disclose the carbon emission report of enterprises in a timely manner and supervise whether or not enterprises have misreported carbon emissions. An excessively high carbon tax rate will reduce the incentive for enterprises to reduce emissions honestly, and the government should set the corresponding carbon tax rate by observing enterprises’ emission reduction behavior. The government should also strengthen the publicity of green finance knowledge, advocate the implementation of the low-carbon economy by enterprises, and control the emission reduction behavior of enterprises by setting dynamic rewards and punishments.(2)The alliance, as the directly responsible entity of enterprises, is mainly responsible for allocating carbon quotas to enterprises and giving them technical support for emission reduction in order to reduce their emission reduction costs. Carbon trading can be implemented within the alliance, and enterprises within the alliance enjoy preferential carbon trading prices compared to the market. The alliance should always pay attention to the market dynamics and enterprises’ emission reduction behavior and adjust the carbon trading price within the industry in a timely manner. Appropriate preferential carbon trading price will increase the enthusiasm of enterprises to reduce emissions honestly and ensure the effectiveness of technical support given by the alliance to enterprises. If other places have similar backgrounds to China’s carbon emission reduction market, this model using the enterprise alliance game party for carbon emission reduction may also be applicable elsewhere in the world using similar strategies.(3)Enterprises are encouraged to establish environmental awareness, accelerate technology development and promotion, strengthen the communication and cooperation among enterprises, and work together to achieve a low-carbon economy.

Although the three-party game model constructed in this paper makes a breakthrough in innovating the government supervision mode, stimulating the vitality of enterprise subjects, and promoting industrial technological innovation, the paper does not discuss the situation of the unequal distribution of carbon quotas among enterprises within the alliance. The leading enterprises within the alliance may demand more carbon quotas and government rewards due to the role of technology demonstration and technical support, and this will be a potential challenge to the win-win cooperation behavior of alliance enterprise, which will be the direction of further research in this paper.

## Figures and Tables

**Figure 1 ijerph-19-11368-f001:**
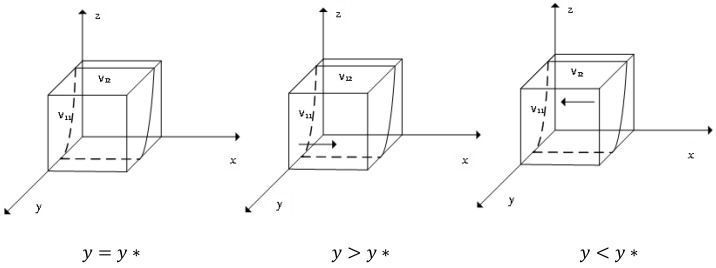
Evolutionary phase diagram of enterprise stabilization strategies.

**Figure 2 ijerph-19-11368-f002:**
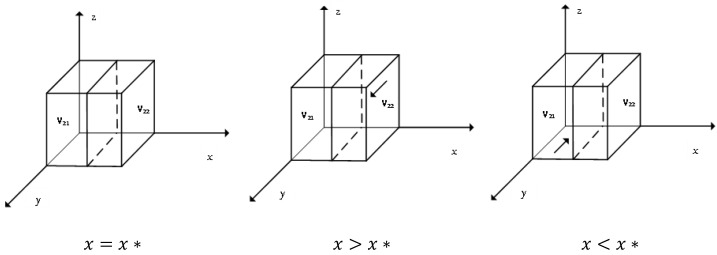
Evolutionary phase diagram of alliance stabilization strategies.

**Figure 3 ijerph-19-11368-f003:**
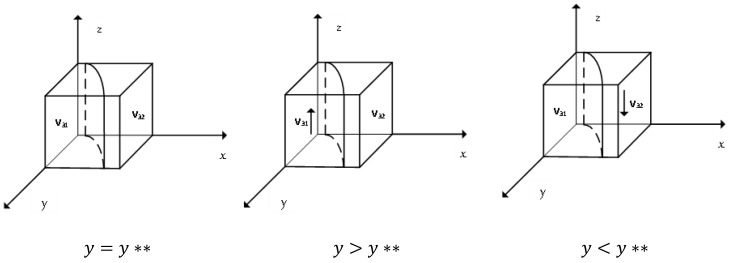
Evolutionary phase diagram of the government stabilization strategies.

**Figure 4 ijerph-19-11368-f004:**
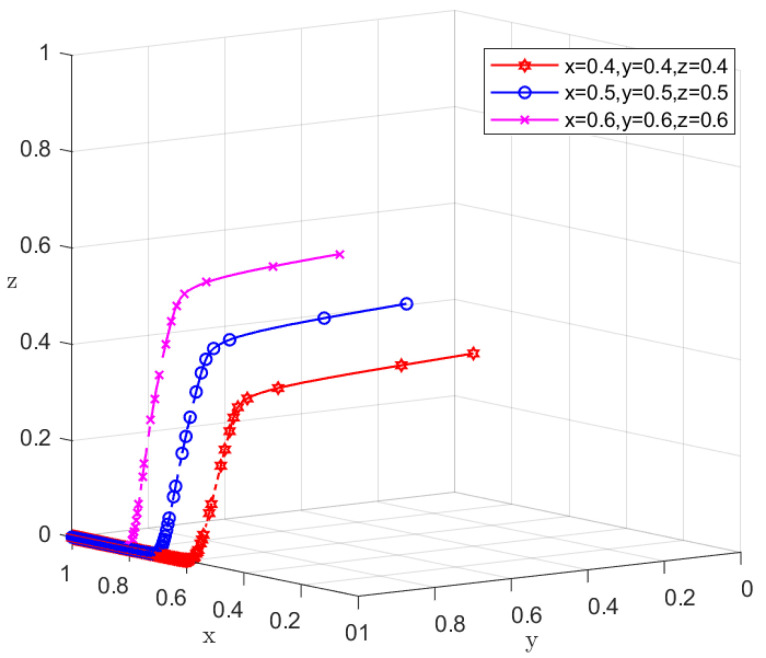
Evolutionary game diagram of three parties under different initial wills.

**Figure 5 ijerph-19-11368-f005:**
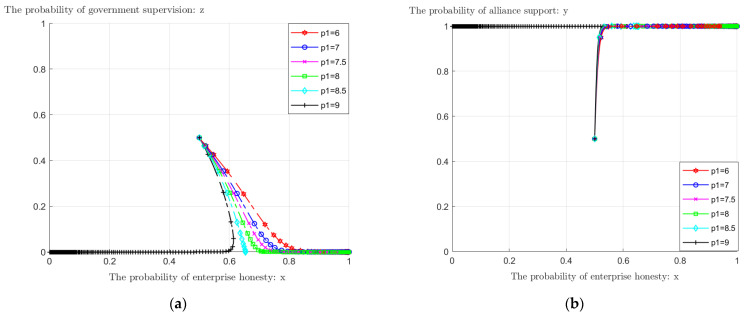
The impact of *P*_1_ change in carbon trading price within the alliance on evolution results: (**a**) impact on the choice of enterprises and government; (**b**) impact on the choice of enterprises and enterprise alliance.

**Figure 6 ijerph-19-11368-f006:**
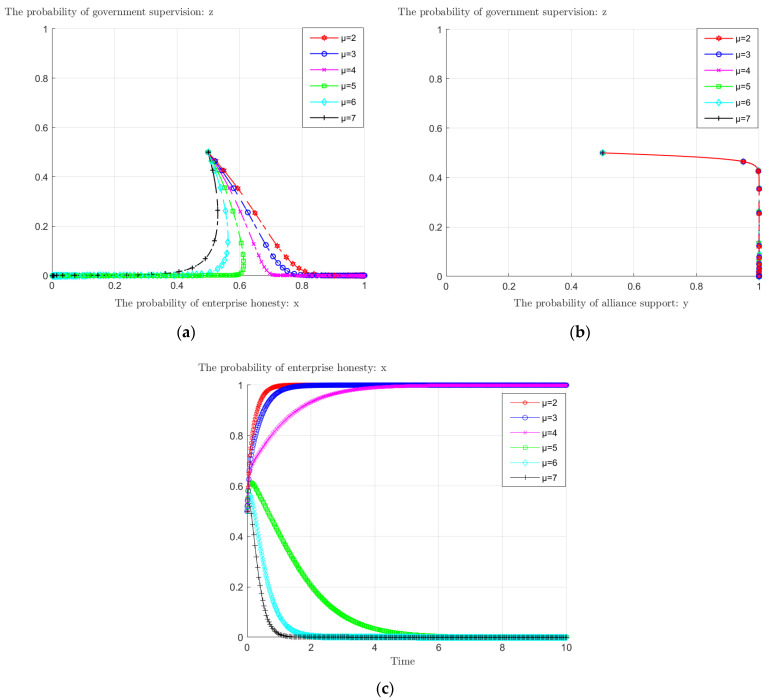
The impact of carbon tax rate *μ* on: (**a**) enterprises’ and governmental behavioral evolution; (**b**) alliance’s and governmental behavioral evolution; (**c**) enterprises’ behavioral evolution.

**Figure 7 ijerph-19-11368-f007:**
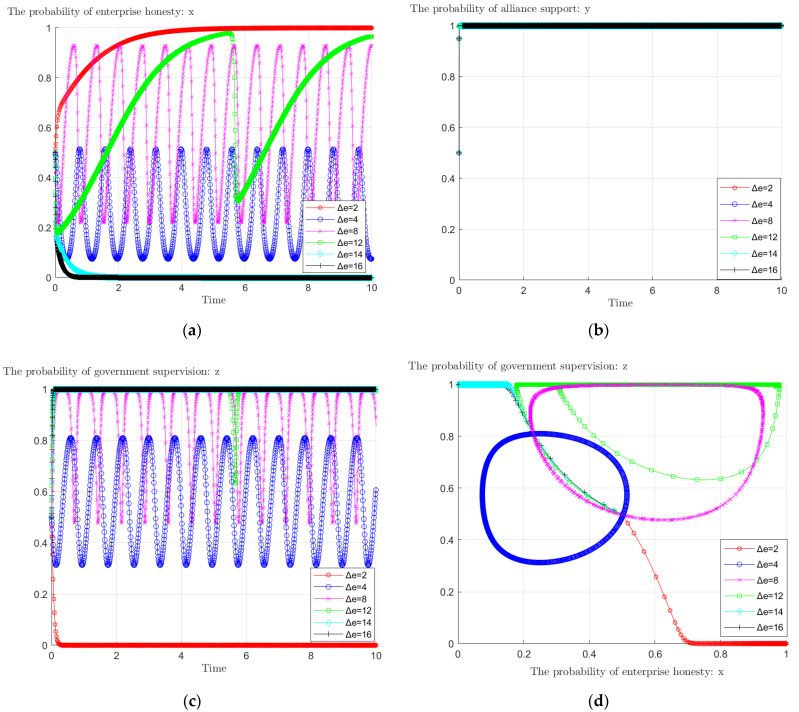
The impact of Δ*e* on: (**a**) enterprises’ behavioral evolution; (**b**) alliance’s behavioral evolution; (**c**) governmental behavioral evolution; (**d**) governmental and enterprises’ behavioral evolution.

**Table 1 ijerph-19-11368-t001:** Parameters’ setting.

Parameters	Descriptions
*R*_1_ (*R*_2_)	The market profit of enterprise honesty (dishonesty)
*C*_1_ (*C*_2_)	The cost of the enterprise honesty when alliance supports (does not support)
*C*_3_ (*C*_4_)	The cost of the enterprise dishonesty when alliance supports (does not support)
*P*_1_ (*P*_2_)	Carbon trading price of alliance support (does not support)
*e* _0_	Free carbon quota for enterprise
*e*_1_ (*e*_2_)	Carbon emissions of enterprise honesty when alliance supports (does not support)
*e*_3_ (*e*_4_)	Carbon emissions of enterprise dishonesty when alliance supports (does not support)
Δ*e*	Underreported carbon emissions by enterprises (Δ*e = e*_1_ *− e*_3_)
*µ*	Carbon tax rate levied on enterprises
*J*	Awards given by the government for technical support of the alliance
*α*	The proportion of technical support awarded by the alliance
*γ*	Penalty coefficient for dishonest emission reduction of enterprises
*C* _0_	The cost of alliance technical support
*S*_1_ (*S*_3_)	The benefits of the alliance support (does not support) when the enterprise is honest
*S*_2_ (*S*_4_)	The benefits of the alliance support (does not support) when the enterprise is dishonest
*N*	Number of enterprises under the alliance
*β*	The reward and punishment coefficient of the government to the alliance
*C*	The cost of government supervision
*H*_1_ (*H*_2_)	Green benefits of the government of the alliance technology support (does not support)
*L*	Environmental losses caused by dishonest emission reduction of enterprises

**Table 2 ijerph-19-11368-t002:** Income matrix of government, enterprises and alliance.

Game Players				Government	
				Supervision	Non-Supervision
Enterprise alliance	Support	enterprise	Honesty	*R*_1_ − *C*_1_ − *P*_1_(*e*_1_ − *e*_0_) − *µe*_1_ + (1 − *α*)*J*	*R*_1_ − *C*_1_ − *P*_1_(*e*_1_ − *e*_0_) − *µe*_1_ + (1 − *α*)*J*
				*S*_1_ + *Nβ*(*e*_0_ − *e*_1_) + *αJ* − *C*_0_	*S*_1_ + *Nβ*(*e*_0_ − *e*_1_) + *αJ* − *C*_0_
				*H*_1_ + *µe*_1_ − *J* − *Nβ*(*e*_0_ − *e*_1_) − *C*	*H*_1_ + *µe*_1_ − *J* − *Nβ*(*e*_0_ − *e*_1_)
			Dishonesty	*R*_2_ − *C*_3_ − *P*_1_(*e*_3_ − *e*_0_) − *µe*_3_ + (1 − *α*)*J* − *γ*(*e*_1_ − *e*_3_)	*R*_2_ − *C*_3_ − *P*_1_(*e*_3_ − *e*_0_) − *µe*_3_ + (1 − *α*)*J*
				*S*_2_ + *Nβ*(*e*_0_ − *e*_3_) + *αJ* − *C*_0_	*S*_2_ + *Nβ*(*e*_0_ − *e*_3_) + *αJ* − *C*_0_
				*H*_1_ + *µe*_3_ − *J* − *Nβ*(*e*_0_ − *e*_3_) − *L* + *γ*(*e*_1_ − *e*_3_) − *C*	*H*_1_ + *µe*_3_ − *J* − *Nβ*(*e*_0_ − *e*_1_) − *L*
	No support	Enterprise	Honesty	*R*_1_ − *C*_2_ − *P*_2_(*e*_2_ − *e*_0_) − *µe*_2_	*R*_1_ − *C*_2_ − *P*_2_(*e*_2_ − *e*_0_) − *µe*_2_
				*S*_3_ + *Nβ*(*e*_0_ − *e*_2_)	*S*_3_ + *Nβ*(*e*_0_ − *e*_2_)
				*H*_2_ + *µe*_2_ − *Nβ*(*e*_0_ − *e*_2_) − *C*	*H*_2_ + *µe*_2_ − *Nβ*(*e*_0_ − *e*_2_)
			Dishonesty	*R*_2_ − *C*_4_ − *P*_2_(*e*_4_ − *e*_0_) − *µe*_4_ − *γ*(*e*_2_ − *e*_4_)	*R*_2_ − *C*_4_ − *P*_2_(*e*_4_ − *e*_0_) − *µe*_4_
				*S*_4_ + *Nβ*(*e*_0_ − *e*_4_)	*S*_4_ + *Nβ*(*e*_0_ − *e*_4_)
				*H*_2_ + *µe*_4_ − *Nβ*(*e*_0_ − *e*_4_) − *L* + *γ*(*e*_2_ − *e*_4_) − *C*	*H*_2_ + *µe*_4_ − *Nβ*(*e*_0_ − *e*_4_) − *L*

**Table 3 ijerph-19-11368-t003:** Eigenvalues of Jacobian matrix for equilibrium points.

Equilibrium Point	Eigenvalue: λ_1_	Eigenvalue: λ_2_	Eigenvalue: λ_3_
(0, 0, 0)	*R*_1_ − *R*_2_ + *C*_4_ − *C*_2_ + *P*_2_(*e*_4_ − *e*_2_) + *µ*(*e*_4_ − *e*_2_)	*αJ* − *C*_0_ + *S*_2_ − *S*_4_ + *Nβ*(*e*_4_ − *e*_3_)	γ(*e*_2_ − *e*_4_) − *C*
(0, 1, 0)	*R*_1_ − *R*_2_ + *C*_3_ − *C*_1_ + *P*_1_(*e*_3_ − *e*_1_) + *µ*(*e*_3_ − *e*_1_)	−[*αJ* − *C*_0_ + *S*_2_ − *S*_4_ + *Nβ*(*e*_4_ − *e*_3_)]	*γ*(*e*_1_ − *e*_3_) − *C*
(0, 0, 1)	*R*_1_ − *R*_2_ + *C*_4_ − *C*_2_ + *P*_2_(*e*_4_ − *e*_2_) + (*µ* − *γ*)(*e*_4_ − *e*_2_)	*αJ* − *C*_0_ + *S*_2_ − *S*_4_ + *Nβ*(*e*_4_ − *e*_3_)	−[*γ*(*e*_2_ − *e*_4_) − *C*]
(0, 1, 1)	*R*_1_ − *R*_2_ + *C*_3_ − *C*_1_ + *P*_1_(*e*_3_ − *e*_1_) + (*µ* − *γ*)(*e*_3_ − *e*_1_)	−[*αJ* − *C*_0_ + *S*_2_ − *S*_4_ + *Nβ*(*e*_4_ − *e*_3_)]	−[*γ*(*e*_1_ − *e*_3_) − *C*]
(1, 0, 0)	−[*R*_1_ − *R*_2_ + *C*_4_ − *C*_2_ + *P*_2_(*e*_4_ − *e*_2_) + *µ*(*e*_4_ − *e*_2_)]	*αJ* − *C*_0_ + *S*_1_ − *S*_3_ + *Nβ*(*e*_2_ − *e*_1_)	−*C*
(1, 1, 0)	−[*R*_1_ − *R*_2_ + *C*_3_ − *C*_1_ + *P*_1_(*e*_3_ − *e*_1_) + *µ*(*e*_3 −_ *e*_1_)]	−[*αJ* − *C*_0_ + *S*_1_ − *S*_3_ + *Nβ*(*e*_2_ − *e*_1_)]	−*C*
(1, 0, 1)	−[*R*_1_ − *R*_2_ + *C*_4_ − *C*_2_ + *P*_2_(*e*_4_ − *e*_2_) + (*µ* − *γ*)(*e*_4_ − *e*_2_)]	*αJ* − *C*_0_ + *S*_1_ − *S*_3_ + *Nβ*(*e*_2_ − *e*_1_)	*C*
(1, 1, 1)	−[*R*_1_ − *R*_2_ + *C*_3_ − *C*_1_ + *P*_1_(*e*_3_ − *e*_1_) + (*µ* − *γ*)(*e*_3_ − *e*_1_)]	−[*αJ* − *C*_0_ + *S*_1_ − *S*_3_ + *Nβ*(*e*_2_ − *e*_1_)]	*C*

**Table 4 ijerph-19-11368-t004:** Stability of each equilibrium point under the three mechanisms.

Equilibrium Point	Carbon Tax, Carbon Trading				Carbon Tax, No Carbon Trading				No Carbon Tax, Carbon Trading			
	Symbols			State	Symbols			State	Symbols			State
(0, 0, 0)	N	+	N	Unstable point	N	+	N	Unstable point	N	+	N	Unstable point
(0, 1, 0)	N	-	N	ESS	N	-	N	ESS	N	-	N	ESS
(0, 0, 1)	N	+	N	Unstable point	N	+	N	Unstable point	N	+	N	Unstable point
(0, 1, 1)	N	-	N	ESS	N	-	N	ESS	N	-	N	ESS
(1, 0, 0)	N	+	-	Unstable point	N	+	-	Unstable point	N	+	-	Unstable point
(1, 1, 0)	N	-	-	ESS	N	-	-	ESS	N	-	-	ESS
(1, 0, 1)	N	+	+	Unstable point	N	+	+	Unstable point	N	+	+	Unstable point
(1, 1, 1)	N	-	+	Unstable point	N	-	+	Unstable point	N	-	+	Unstable point

Note: “+” indicates a positive number; “-” indicates a negative number; “N” indicates uncertainty.

**Table 5 ijerph-19-11368-t005:** Conditions that need to be met for the stabilization point.

ESS	Carbon Tax, Carbon Trading	Carbon Tax, No Carbon Trading	No Carbon Tax, Carbon Trading
(0, 1, 0)	γΔe<C $P_{1}+\mu >\frac{R_{1}-C_{1}-R_{2}+C_{3}}{\Delta e}$	γΔe<C $\mu >\frac{R_{1}-C_{1}-R_{2}+C_{3}}{\Delta e}$	γΔe<C $P_{1}>\frac{R_{1}-C_{1}-R_{2}+C_{3}}{\Delta e}$
(0, 1, 1)	γΔe>C $P_{1}+\mu >\gamma +\frac{R_{1}-C_{1}-R_{2}+C_{3}}{\Delta e}$	γΔe>C $\mu >\gamma +\frac{R_{1}-C_{1}-R_{2}+C_{3}}{\Delta e}$	γΔe>C $P_{1}>\gamma +\frac{R_{1}-C_{1}-R_{2}+C_{3}}{\Delta e}$
(1, 1, 0)	$P_{1}+\mu <\frac{R_{1}-C_{1}-R_{2}+C_{3}}{\Delta e}$	$\mu <\frac{R_{1}-C_{1}-R_{2}+C_{3}}{\Delta e}$	$P_{1}<\frac{R_{1}-C_{1}-R_{2}+C_{3}}{\Delta e}$

## Data Availability

No new data were created or analyzed in this study. Data sharing is not applicable to this article.

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
