# Peer review of "Role of Enterprise Alliance in Carbon Emission Reduction Mechanism: An Evolutionary Game Analysis"

_ijerph, 2022, doi:10.3390/ijerph191811368_

Round 1
Reviewer 1 Report
1. The quality of the paper's figures are very low. Specially Figures 4-7.
2. Kindly enrich literature review with more related papers.
3. Line 555: “propaganda” is not proper word for the sentence and has negative implication.
4. Line 571: “optimizing government regulation” doesn’t seem to be a proper term.
Author Response
Point 1: The quality of the paper's figures are very low. Specially Figures 4-7.
Response 1: Thank you for your valuable comments. We have made adjustments based on your comments. The figures are directly imported from the Matlab software and their resolution is greater than 500 PPI. We believe the clarity of the figures has met the requirements of the journal.
Point 2: Kindly enrich literature review with more related papers.
Response 2: As suggested by the reviewer,we have added more references into the introduction part in the revised manuscript. Specifically, we have added papers 7, 10, 13, and 16 to study the relationship between government and enterprise behavior under the emission reduction mechanisms. We have added papers 25 to study how to achieve a win-win situation for multiple players. We have added papers 30, 31, and 32 to study how to innovate low-carbon technologies for enterprises.
Point 3: Line 555: “propaganda” is not proper word for the sentence and has negative implication.
Response 3: We were really sorry for our careless mistake. Thank you for your reminder. We learned that the word propaganda has negative implication, and changed “propaganda” to “publicity”.
Point 4: Line 571: “optimizing government regulation” doesn’t seem to be a proper term.
Response 4: Thank you for pointing this out. Our expression is wrong, what we want to express is to change the original way of government regulation. We have corrected the “optimizing government regulation” into “innovating the government supervision mode”.
Reviewer 2 Report
The article “Role of Enterprise Alliance in Carbon Emission Reduction Mechanism: An Evolutionary Game Analysis” descries the construction of an evolutionary game model based on three partners (government, enterprises and enterprises alliance) considering different mechanisms for carbon emission reduction. Strategies with and without carbon trading and carbon trading mechanisms are studied considering three different emission reduction mechanisms for finding the optimal path of carbon emission reduction. Numerical experiments and simulations are carried out to validate the developed evolutionary game model to investigate the impact on the key variables and the strategies of the model. Finally conclusions and related policy recommendations are formulated.
The article is well-written and documented with interesting outcomes and results.
However there are a series of issues which require more explanation.
1. In the introduction it is stated that China is “clearly emphasized the need to vigorously build a low-carbon recycling development economic system and reduce the level of carbon dioxide emissions. Is this model using the enterprise alliance game party for carbon emission reduction also applicable elsewhere in the world using similar mechanisms and/or strategies? Is the outcome of this evolutionary game model based on three partners dependent on the mechanisms proposed by the government for reducing CO2 emissions (line 43)? Please comment.
2. Is there an optimum or a restriction in the number (N) of enterprises participating in the enterprise alliance? Can there be more than one alliance e.g. Alliance of small and medium–sized companies and alliance of large enterprises as third party? Please comment.
3. In Table 2 ”J” (Jacobian matrix?) is not explained nor found in Table 1. Please comment.
4. Line 240: “the expected profit when it chooses to reduce emissions honestly is denoted as ?12” should be “the expected profit when it chooses not to reduce emissions honestly is denoted as ?12“. Please comment.
5. Lines 418-421: editing mode should be adapted.
6. Figures 5 and corresponding labels are not clearly presented and should be adapted.
7. Figures 6 and corresponding labels are not clearly presented and should be adapted.
8. Figures 7 and corresponding labels are not clearly presented and should be adapted.
9. Line 52: ”industrial” instead of “industry”
Author Response
Point 1: In the introduction it is stated that China is “clearly emphasized the need to vigorously build a low-carbon recycling development economic system and reduce the level of carbon dioxide emissions. Is this model using the enterprise alliance game party for carbon emission reduction also applicable elsewhere in the world using similar mechanisms and/or strategies? Is the outcome of this evolutionary game model based on three partners dependent on the mechanisms proposed by the government for reducing CO2 emissions (line 43)? Please comment.
Response 1: 1) Thank you for raising this valuable question, and we have made a serious consideration. Based on the relationship between enterprises and the government in China's emission reduction market, this paper constructs an enterprise alliance mechanism. In this market, enterprises have dishonest emission reduction behaviors due to insufficient carbon reduction capacity and difficulties for the government to check carbon emission reports. The government actively regulates enterprise behaviors through a series of emission reduction policies. If other places have similar backgrounds to China's carbon emission reduction market,this model using the enterprise alliance game party for carbon emission reduction also applicable elsewhere in the world using similar strategies. We have also added this note in the conclusion section of the revised manuscript.
2) Yes, enterprises, enterprise alliance and the government influence each other, and the government's emission reduction mechanisms are important factors influencing enterprises' choices. Among them, literature 5 shows that appropriate reward and punishment mechanisms will motivate enterprises to reduce emissions, and literature 6 shows that rewards are not conducive to the government to fulfill its regulatory responsibilities. Carbon trading policy can effectively curb the carbon emissions of enterprises, as confirmed by literature 10. And the high rate of carbon tax set by the government can lead to dishonest emission reduction by enterprises. Therefore, it is said that the outcome of this evolutionary game model dependent on the mechanisms proposed by the government for reducing CO2 emissions.
Point 2: Is there an optimum or a restriction in the number (N) of enterprises participating in the enterprise alliance? Can there be more than one alliance e.g. Alliance of small and medium–sized companies and alliance of large enterprises as third party? Please comment.
Response 2: We thank the reviewer for raising this question. We think that there is an optimum in the number (N) of enterprises participating in the enterprise alliance. At the beginning, the number of enterprises participating in the alliance was small, but as the leading enterprises within the alliance drive the small enterprises to improve their carbon reduction technology, more and more small and medium-sized enterprises want to join the alliance, which is not a good benefit for the leading enterprises if the carbon quotas allocated to large and small enterprises are the same. Therefore, it is possible to adjust the carbon quotas of enterprises and the number of participating alliance to maximize the win-win cooperation of multiple players. At the same time, we also think that there can be more than one alliance e.g. Alliance of small and medium–sized companies and alliance of large enterprises as third party. It is a pity that this paper has not studied this content. I think these problems can be taken as the next research direction.
Point 3: In Table 2 ”J” (Jacobian matrix?) is not explained nor found in Table 1. Please comment.
Response 3: We feel sorry for our carelessness. “J” in Table 2 is not a Jacobian matrix. “J” is the reward given by the government to the enterprise alliance when the enterprise alliance chooses to provide technical support to the enterprise. "J" is explained in Table 2, but we carelessly marked “J” as “i”, and we have modified it in Table 2. In addition, “J” is also explained in hypothesis 2 and hypothesis 3 in the second part of the article.
Point 4: Line 240: “the expected profit when it chooses to reduce emissions honestly is denoted as ?12” should be “the expected profit when it chooses not to reduce emissions honestly is denoted as ?12“. Please comment.
Response 4: Thanks for your careful checks. We have made the appropriate correction.
Point 5: Lines 418-421: editing mode should be adapted.
Response 5:, I don't know if there is a problem opening the format, but lines 418-421 show the last line of Table 5 and the fourth part of the title. We have carefully checked the manuscript and adjusted the formatting around lines 418-421 to make the table look better, changing the outline level of some lines from "first level headings" to the correct "body text".
Point 6: Figures 5 and corresponding labels are not clearly presented and should be adapted.
Response 6: Thank you for your comments. We have made adjustments to Figure 5 and the corresponding labels.
Point 7: Figures 6 and corresponding labels are not clearly presented and should be adapted.
Response 7: Thank you for your comments. We have made adjustments to Figure 6 and the corresponding labels.
Point 8: Figures 7 and corresponding labels are not clearly presented and should be adapted.
Response 8: Thank you for your comments. We have made adjustments to Figure 7 and the corresponding labels.
Point 9: Line 52: ”industrial” instead of “industry”
Response 9: Thank you for pointing this out. Although we did not see the word “industry” in line 52. However, we looked up and checked all the words in the article for “industry” and found two errors, we have corrected the “industry technological innovation” into “industrial technological innovation”.
Reviewer 3 Report
This paper creates an evolutionary game model consisting of three parties: the government, enterprises, and a new player, called "Enterprise Alliance". Enterprise Alliance has the goal of reducing total emissions of its member enterprises, facilitating technical innovation to achieve this, and manages the use of the carbon quota for enterprises in the alliance. The paper investigates how players' evolutionarily stable strategies are impacted by the carbon trading price, carbon tax rate and corporate dishonesty (in the form of underreporting of carbon emissions).
Overall this is a well-constructed paper with a worthwhile outcome and provides useful direction for policymakers. The following elements need addressing: 1. The paper is mostly well-written, but there are a few points where the English language may be improved (see, for example, line 44, lines 121-122 and line 125). 2. The Author contributions and Acknowledgments sections have not been filled out.
Author Response
Point 1: The paper is mostly well-written, but there are a few points where the English language may be improved (see, for example, line 44, lines 121-122 and line 125).
Response 1: Thank you very much for your approval of our article and for raising the English language issue. We have corrected your comment about the need for improvement and other areas of improvement and marked them with a revision pattern.
Point 2: The Author contributions and Acknowledgments sections have not been filled out.
Response 2: Thank you for your reminder. We have filled in the Author contributions and Acknowledged sections.
Reviewer 4 Report
The manuscript presents the construction of an evolutionary game model where three game players among enterprises, enterprise alliance and the government have been involved, considering also the mechanisms about carbon tax, carbon trading, government reward and punishment.
The evolutionary game analysis has been than conducted and explained in detail and the defined model has been validated using numerical simulation.
The manuscript is well written, clear and uses standard nomenclature to describe both model and numerical simulation details.
Also the introduction is well written and contains the needed information about the related work.
The conclusions recaps the content pf the paper by highlighting the policy implications deduced by the analysis of the outcomes.
Minor spell check are required and the alignement of the figures in respect to the text should be checked.
Author Response
Point: Minor spell check are required and the alignement of the figures in respect to the text should be checked.
Response: First of all, thank you very much for your approval of this article. We have carefully spell-checked the manuscript and aligned the figures with the text, made the appropriate changes and marked the changes in red using the revision function.
Round 2
Reviewer 3 Report
The authors have responded very well to reviewers' comments and also have made relevant revisions.